# Deep Learning Evaluation of Glaucoma Detection Using Fundus Photographs in Highly Myopic Populations

**DOI:** 10.3390/biomedicines12071394

**Published:** 2024-06-23

**Authors:** Yen-Ying Chiang, Ching-Long Chen, Yi-Hao Chen

**Affiliations:** 1Graduate Institute of Life Sciences, National Defense Medical Center, Taipei 114, Taiwan; h200345@hotmail.com; 2Department of Ophthalmology, Tri-Service General Hospital, National Defense Medical Center, Taipei 114, Taiwan; doc30881@mail.ndmctsgh.edu.tw

**Keywords:** glaucoma, myopia, artificial intelligence, fundus photographs

## Abstract

Objectives: This study aimed to use deep learning to identify glaucoma and normal eyes in groups with high myopia using fundus photographs. Methods: Patients who visited Tri-Services General Hospital from 1 November 2018 to 31 October 2022 were retrospectively reviewed. Patients with high myopia (spherical equivalent refraction of ≤−6.0 D) were included in the current analysis. Meanwhile, patients with pathological myopia were excluded. The participants were then divided into the high myopia group and high myopia glaucoma group. We used two classification models with the convolutional block attention module (CBAM), an attention mechanism module that enhances the performance of convolutional neural networks (CNNs), to investigate glaucoma cases. The learning data of this experiment were evaluated through fivefold cross-validation. The images were categorized into training, validation, and test sets in a ratio of 6:2:2. Grad-CAM visual visualization improved the interpretability of the CNN results. The performance indicators for evaluating the model include the area under the receiver operating characteristic curve (AUC), sensitivity, and specificity. Results: A total of 3088 fundus photographs were used for the deep-learning model, including 1540 and 1548 fundus photographs for the high myopia glaucoma and high myopia groups, respectively. The average refractive power of the high myopia glaucoma group and the high myopia group were −8.83 ± 2.9 D and −8.73 ± 2.6 D, respectively (*p* = 0.30). Based on a fivefold cross-validation assessment, the ConvNeXt_Base+CBAM architecture had the best performance, with an AUC of 0.894, accuracy of 82.16%, sensitivity of 81.04%, specificity of 83.27%, and F1 score of 81.92%. Conclusions: Glaucoma in individuals with high myopia was identified from their fundus photographs.

## 1. Introduction

In recent years, the prevalence of myopia has increased rapidly particularly in East and Southeast Asian countries, such as Singapore, China, Taiwan, Hong Kong, Japan, and South Korea [1]. Based on surveys, since 2015 [2], approximately 1.406 billion people (22.9% of the total population) globally can develop myopia. Further, approximately 163 million people (2.7% of the total population) have high myopia (spherical equivalent refraction [SER] of <−5.0 D). According to the development trend of high myopia, considering environmental and lifestyle changes, the incidence of high myopia worldwide can reach approximately 10% by 2050. Thus, it is a public health issue that cannot be ignored. Glaucoma is a progressive optic neuropathy [3]. Injured optic nerve causing visual field loss is commonly irreversible [4,5]. Globally, >76 million people are diagnosed with glaucoma, which reduces the quality of life, impairs vision, and has an increasing incidence [6,7]. The severity of visual field defects will further increase with age [8,9]. Patients with glaucoma require long-term treatment and examination to avoid continued degradation and loss of visual field. Myopia is an important risk factor for glaucoma. The risk of developing glaucoma increases alongside myopia severity [10,11]. A meta-analysis study by Ahnul Ha et al. included seven studies reporting risk estimates for high myopia and revealed that the pooled odds ratio for developing glaucoma was 4.142 (95% confidence interval [CI]: 2.567–6.685) [12]. As the population of individuals with high myopia increases, it is extremely important to accurately diagnose glaucoma. However, clinicians often encounter challenges when diagnosing patients with glaucoma who present with high myopia. This is caused by the structure and function similarities between glaucoma and myopia [13].

Axial elongation because of high myopia changes the retinal structure and function, including morphology similar to that seen in patients with glaucoma, such as disc size, shape, neuroretinal rim shape, and pallor [14,15]. As high myopia worsens, the axial length of the eye elongates, and the optic disc stretches horizontally into an oval shape on fundus photography. Further, the condition may cause sagittal rotation of the optic disc, which is referred to as “tilted disc”. The normal optic disc shape and glaucoma are vertically elliptical. The cup-to-disc ratio in nonglaucoma high myopia is approximately 1.5 times higher than that without myopia [16]. However, previous research has shown that the optic disc size is not an important risk factor for the development of glaucoma [17]. In myopia, the height between the neuroretinal edge and the base of the optic cup is reduced by flattening the optic cup. Simultaneously, the Bruch’s membrane opening expands, and the retinal edge becomes thinner. The neuroretinal edge loss in glaucoma progresses as inferotemporal, supratemporal, infranasal, and supranasal as the disease progresses [18,19,20]. High myopic glaucoma is more difficult to diagnose because of the abnormal shape of the optic retinal edge, which no longer conforms to the inferior–superior–nasal–temporal rule [21,22]. High myopia and glaucoma cause thinning of the retinal nerve fiber layer. High myopia is mainly caused by the excessive elongation of the eyeball, and RNFL thinning is mainly observed in the superior and inferior temporal regions. Meanwhile, glaucoma is attributed to optic nerve damage caused by increased intraocular pressure. Further, the thickness of the RNFL and ganglion cell complex decreases significantly over time [23,24]. According to the progression of visual field defects in myopia and glaucoma, non-myopic glaucoma usually manifests as Bjerrum area defects and nasal steps in the early stage. Further, central visual field defects develop over time. In the early stage of myopic glaucoma, central or paracentral darkening is more common [25]. High myopia can be interpreted as false-positive glaucoma when diagnosing glaucoma [26,27]. Fundus photography, optical coherence tomography (OCT) is usually required to compare the results of long-term tracking changes with those of visual field examination. Therefore, diagnosing glaucoma in highly myopic eyes is difficult.

In recent years, artificial intelligence has made significant progress in the field of medical imaging. In particular, the application of convolutional neural networks (CNNs) is extremely suitable for processing spatial patterns and performing tasks, such as image classification and object detection, thereby promoting the development of deep-learning technology for image classification and pattern recognition [28]. Deep-learning algorithms require a large amount of data for training, which is particularly challenging when clinical data are limited. At this point, transfer learning can often leverage models pretrained on the dataset, thereby reducing training time and improving performance [29]. Several ophthalmologic diseases, such as diabetic retinopathy [30], glaucoma [31,32], macular degeneration [33], and myopia [34], have been evaluated via artificial intelligence medical imaging. The artificial intelligence-assisted diagnosis of glaucoma mainly involves analyzing OCT, visual field test results, and fundus photography. Fundus photography has significant advantages in related applications as they are accessible, have a high quality, and are cost-effective [35].

In the study by Li et al. [36], ophthalmologists classified 48,116 fundus photographs. Glaucoma was defined as a vertical cup-to-disc ratio of ≥0.7. In the Inception-v3 architecture, the AUC was 0.986, the accuracy was 92.9%, the sensitivity was 95.6%, and the specificity was 92.0%. In the study by Kim et al. [37], 747 myopic healthy eyes and 1860 myopic glaucoma eyes were included, using macular vertical OCT to evaluate the glaucoma diagnostic ability of patients with high myopia. In the EfficientNet architecture, the external test dataset showed an AUC of 0.984 using macular vertical OCT scans and an AUC of 0.983 using individual data combinations. Asaoka et al. analyzed the visual field of the first diagnosis of evident glaucoma and the visual field of healthy people. Deep learning can be used to predict the development trend of glaucoma and achieve good outcomes (AUC: 92.6%) [38]. Regarding fundus photography medical imaging detection of glaucoma, previous studies focused more on the cutting of the optic disc and the structure of the optic nerve head, although they all have excellent performance [38,39,40,41,42]. However, fundus photos with a viewing angle of 45° are not always used, and other signs in the eye map may be ignored.

Previous deep-learning studies have used non-highly myopic or OCT parameters for deep learning, or overly focused on the optic nerve head structure. Moreover, currently, no study has utilized fundus photos for evaluating high myopia glaucoma. Herein, several novel issues were explored. First, this is the study to specifically use fundus photographs for detecting glaucoma in patients with high myopia. Second, we develop a deep-learning framework incorporating convolutional block attention modules (CBAMs) to enhance the performance of CNN glaucoma detection. Finally, the use of Grad-CAM visualization improves the interpretability of CNN results and helps comprehend how the model makes predictions. The use of fundus photographs can prevent issues related to the inability to convert data between different machines such as OCT. The fundus photographs are also available easily and are time-saving [39,40]. This study aimed to use deep learning to detect glaucoma on the fundus photographs of individuals with high myopia. We believe that this method can assist ophthalmologists in achieving an accurate diagnosis or other professionals, such as optometrists, family physicians, and non-ophthalmologists, in making early referrals.

## 2. Materials and Methods

### 2.1. Ethics, Consent, and Permissions

This study obtained ethical approval from the research ethics committee of the Tri-Service General Hospital (TSGHIRB No. C202305105) and adhered to the principles of the Declaration of Helsinki.

### 2.2. Datasets Collection

We retrospectively reviewed patients who visited Tri-Services General Hospital from 1 November 2018 to 31 October 2022. Initially, there were 35,327 patients with 126,955 fundus photographs. All patients underwent visual acuity assessment, refractive error, and slit-lamp biomicroscopy. All patients with high myopia underwent non-mydriatic fundus photography (AFC-330; Nideck Co. Ltd., Gamagori, Aichi, Japan), whereas all patients with glaucoma underwent central 24-2 threshold testing using a standard Humphrey Field Analyzer (Carl Zeiss Meditec, Dublin, CA, USA) and optical coherence tomography (RTVue-100; Optovue Inc., Fremont, CA, USA). OCT was used to measure optic disc cupping, circumpapillary retinal nerve fiber layer thickness, and macular layer thickness. All clinical records were deidentified and anonymized before analysis in this study.

The inclusion criteria for people with high myopia were (1) age of >20 years at baseline examination and (2) SER of ≤−6.0 D [41]. SER is a measure of the refractive error of the eye and is calculated using the following formula: spherical power + 0.5 * (cylindrical power). This study excludes patients with pathological myopia, those who have undergone cataract surgery, refractive surgery, diabetes, and macular degeneration, any other ocular or systemic disease affecting the retinal nerve fiber layer, and other diseases impacting the visual field (e.g., neuroophthalmic disease, uveitis, retinal, or choroidal disease, trauma), as well as those with severe media opacity that interferes with fundus photography or OCT image acquisition. High myopia includes axial myopia and refractive myopia, and the axial length of the eye is not used as a single inclusion criterion.

Patients with glaucoma were selected after excluding those who did not meet the criteria. All patients with glaucoma were diagnosed after long-term follow-up by professional ophthalmologists. The inclusion criteria for patients with glaucoma were as follows: first, changes that indicate glaucomatous optic nerve or nerve fiber layer defects, such as increased narrowing of the neuroretinal rim (change in a sector of the neuroretinal rim from narrow to complete loss or from a homogeneous neuroretinal rim to a narrow sector) and a significant expansion of a retinal nerve fiber layer defect on OCT. Second, automated visual field testing showed glaucomatous visual field progression defined using the Anderson–Patella criteria, i.e., the Glaucoma Hemifield Test (GHT) result was outside normal limits. There was a cluster of three or more non-edge points in the typical location of the glaucoma, all of which were depressed on the pattern deviation plot at a *p* value of <5%, with at least one of these points depressed at a *p* value of <1%. In addition, the corrected pattern standard deviation was abnormal at a *p* value of <5%. All patients with glaucoma are receiving antiglaucoma treatment, such as the following eye drops: beta blockers, alpha-2 adrenergic agonists, prostaglandin analogs, and carbonic anhydrase inhibitors. Finally, they were categorized into high myopia and high myopia with glaucoma groups. Different image categories are described in the Appendix A. Figure 1 shows the Flowchart of patient inclusion and exclusion.

### 2.3. Model Building

A CNN retrieves image features for classification tasks through its deep structure, which greatly improves the accuracy and efficiency of disease diagnosis in medical image classification [42]. Our study used two CNN architecture models: EfficientNet_V2-S [43], which was proposed by Tan and Le in 2021, and ConvNeXt_Base [44], which was introduced by Liu et al. in 2022. Additionally, we used transfer learning. Previous literature has revealed that pretrained parameters improve classification capabilities more quickly. The weights are derived from pretraining on the ImageNet visual recognition challenge dataset [45]. The last layer of the model (the fully connected layer) is replaced, and the output is ultimately classified into two categories. The training setup for all model architectures features a batch size of 8 across 30 epochs. Initially, the learning rate starts from 0.01 and decays by 0.3 times every 6 epochs. This configuration used the Cross-Entropy Loss function and the AdamW [46] optimizer with a learning rate of 10^−5^ and weight decay of 10^−5^. Additionally, it includes a StepLR scheduler with a step size of 10 and incorporates a dropout rate of 0.2.

The CBAM (Figure 2) is used in the model to improve the performance of the CNN to improve the accuracy of target detection and object classification. The attention mechanism module [47] CBAM sequentially applies channel and spatial attention mechanisms to progressively refine the attention distribution of feature maps.

The channel attention mechanism assigns a weight to each channel, emphasizing important channels and suppressing less important ones. This is expressed as follows:
(1)
$$M_{c}(F)=\sigma (MLP(AvgPool\;(F))+MLP(MaxPool(F))$$



F
 denotes the input feature map. 
AvgPool
 and 
MaxPool
 indicate the average and max pooling operations across the channel dimension, which capture the channels’ global statistical information. 
MLP
 stands for multilayer perceptron. The symbol 
σ
 signifies the sigmoid activation function, which produces the channel attention weights 
$M_{c}$
.

The spatial attention mechanism emphasizes important spatial regions within the feature maps. This is expressed as follows:
(2)
$$M_{s}\left(F\right)=\sigma \left(\;f^{7\times 7}\;\left(\left[AvgPool\left(F\right);\;MaxPool\left(F\right)\right]\right)\right)$$



F
 indicates the input feature map. First, the feature map independently undergoes average and max pooling across each channel, followed by stacking the two pooling results along the channel dimension. 
$f^{7\times 7}$
 represents a convolution operation with a 
7×7
 kernel, capturing contextual information in the spatial domain. Finally, the sigmoid function 
σ
 generates the spatial attention weights 
$M_{s}$
.

### 2.4. Image Preprocessing

We evaluated the learning data for this experiment through fivefold cross-validation. The dataset is randomly divided into five equal subset sizes in the fivefold cross-validation, and the images are categorized into training, validation, and test sets in a ratio of 6:2:2 (Figure 3). Fundus photographs of each patient are not repeated. Each image will appear in only one of the subsets and will not be repeated. In particular, it only exists in the training set, but not in the validation or test sets. The image was resized to 3 × 224 × 224 and augmented with data, including horizontal flipping and random rotation within 20°. The execution system uses an Intel Core i7-11370H 3.3 GHz (TSMC, Hisnchu, Taiwan) processor with 40 GB memory and an NVIDIA GeForce RTX-3070 with 8 GB DDR6 (Asus, Taipei, Taiwan) distinct graphics card. The implementation of the deep neural network is based on the PyTorch platform version 2.1.0+cu121.

### 2.5. Visualization of the Imaging Features

Heatmap visualization using Grad-CAM [48] improves the comprehensibility of CNN outcomes. Every image produces heat maps that highlight the areas of focus for the deep-learning model. This method involves calculating the derivative of the CNN architecture’s final convolutional layer output relative to each pixel of the input image. Pixels in the image with a higher influence appear nearer to the red spectrum on the heatmap, whereas those with a lesser influence are aligned closer to the blue spectrum.

### 2.6. Statistical Methods

Statistical Package for the Social Sciences version 22.0 (Chicago, IL, USA) was used for statistical analyses. Demographic comparisons between patients with high myopia group and those with high myopia glaucoma group were performed using independent *t*-tests or chi-square tests. *p* values of 0.05 were considered statistically significant. The metrics used to assess the model’s efficacy encompass the area under the curve (AUC) of the receiver operating characteristic (ROC), accuracy, sensitivity, specificity, and F1 score. The ROC curve depicts the balance between sensitivity and the complement of the false positive rate (1—specificity). The AUC under the ROC curve was computed. An AUC value of 1.0 signifies flawless differentiation, whereas a value of 0.5 indicates discrimination equivalent to random chance. Accuracy is a metric that measures the proportion of correctly predicted instances among the total instances in a classification model. The F1 score is a metric that combines precision and sensitivity to evaluate the performance of a classification model, with values ranging from 0 to 1, where a higher score indicates a better model performance. The results of the fivefold cross-validation are averaged to assess generalizability.

## 3. Results

This study initially included 35,327 patients with 126,955 fundus photographs and 16,423 images with high myopia. After excluding patients who did not meet the conditions (*n* = 12,913) and those with poor image quality (*n* = 422), 1637 met the criteria. In total, there were 3088 fundus images, including those of 796 patients in the high myopia glaucoma group, with 1540 fundus images. Women accounted for 48.77% and 59.04% in the high myopia glaucoma and high myopia groups, respectively, in terms of gender (*p* < 0.001). The average age was 47.6 ± 12.0 and 46.4 ± 14.2 (*p* = 0.01). The average diopter was −8.83 ± 2.9 D and −8.73 ± 2.6 D (*p* = 0.30) (Table 1). The OCT parameters of the high myopia glaucoma group were as follows: average RNFL, 78.17 ± 15.2 µm; rim area, 0.9 ± 1.4 mm^2^; disc area, 2.20 ± 2.9 mm^2^; cup volume, 0.38 ± 0.38 mm^3^; and average C/D area ratio, 0.71 ± 3.2. The VF parameters of the high myopia glaucoma group were as follows: average MD, −6.15 ± 7.5 dB and average VFI, 84.88% ± 22.6% (Table 1). Table 2 shows the number of images in the training, validation, and test sets in the fivefold cross-validation.

Table 3 presents the classification performance of the various learning networks. The two different model architectures, EfficientNet_V2-S and ConvNeXt_Base, demonstrated AUCs of 0.86 and 0.870, accuracy of 79.34% and 78.85%, sensitivities of 79.22% and 74.67%, and specificities of 79.46% and 83.01%, F1 score of 79.27% and 77.89%, respectively. The results of using CBAM in the model architecture (EfficientNet_V2-S+CBAM and ConvNeXt_Base+CBAM) indicated AUCs of 0.885 and 0.894, accuracy of 81.38% and 82.16%, sensitivities of 76.56% and 81.04%, and specificities of 86.18% and 83.27%, F1 score of 80.40% and 81.92%, respectively. The results after using CBAM in the model architecture and adding patient characteristics (EfficientNet_V2-S+CBAM+Meta and ConvNeXt_Base+CBAM+Meta) showed AUCs of 0.879 and 0.893, accuracy of 80.38% and 81.77%, sensitivities of 77.62% and 77.60%, and specificities of 84.11% and 85.95%, F1 score of 79.57% and 80.93%, respectively. Figure 4 illustrates the ROC curves and AUC values of various learning networks.

Figure 5 shows the confusion matrix of each classifier under different CNN architectures. The confusion matrices of EfficientNet_V2-S and ConvNeXt_Base were as follows: TP = 1220, 1150 images; FN = 320, 390 images; FP = 318, 263 images; and TN = 1230, 1285 images, respectively. The confusion matrices of EfficientNet_V2-S+CBAM and ConvNeXt_Base+CBAM were as follows: TP = 1179, 1248 images; FN = 361, 292 images; FP = 214, 259 images; and TN = 1334, 1289 images, respectively. The coefficient matrices of EfficientNet_V2-S+CBAM+Meta and ConvNeXt_Base+CBAM+Meta were as follows: TP = 1180, 1195 images; FN = 360, 345 images; FP = 246, 218 images; and TN = 1302, 1330 images, respectively. Figure 6 shows the visual image heat map under the ConvNeXt_Base+CBAM architecture. The upper row (a, b) shows high myopia and the lower row (c, d) exhibits high myopia glaucoma.

## 4. Discussion

This study investigates glaucoma through fundus photographs of highly myopic individuals and reveals good results in CNN deep learning. Glaucoma is the second leading cause of blindness globally after cataracts [49,50]. Screening for glaucoma is essential. OCT and visual field examination require ample time; thus, our study uses deep learning to provide a simple way to screen for glaucoma. This study indicates two differences. First, the use of fivefold cross-validation does not only use the model performance validation for a single test set, so the generalization ability and reliability of the model can be effectively improved during the training process. Second, we placed the CBAM behind the image input layer and in front of the fully connected layer, two positions used to enhance features. EfficientNet_V2-S and ConvNeXt_Base use ImageNet pretrained weights, with AUCs of 0.861 and 0.870, respectively. Table 3 shows that ConvNeXt_Base demonstrated a better performance. The use of CBAM demonstrated the AUC values of 0.885 and 0.894 in the EfficientNet_V2-S+CBAM and ConvNeXt_Base+CBAM models, respectively, indicating that CBAM improves model accuracy and exhibits good performance, which is consistent with previous research results [47]. EfficientNet_V2-S+CBAM+Meta and ConvNeXt_Base+CBAM+Meta added with more patient characteristics, including gender, age, and diopter, demonstrated AUC values of 0.879 and 0.893, respectively. Surprisingly, model performance decreased after adding patient characteristics. A significant difference in the male-to-female ratio was found between the high myopia group and the high myopia glaucoma group (*p* < 0.001), but the model performance did not improve after adding gender characteristics, and it did not affect the model performance in this study.

Although the incidence of high myopia is low, deep learning requires a large amount of data to improve performance of the model. Hence, pretraining was used, and it was combined with CBAM. Based on previous research [51,52,53,54], we utilized CNN models with different architectures, such as EfficientNet_V2-S, ConvNeXt_Base, ResNet50, and ViT_B_16. The model with a higher accuracy in ImageNet Top-1 image classification was used as a basis for the initial training. Finally, EfficientNet_V2-S and ConvNeXt_Base, which have a better performance, were selected as the final model architecture. Among the models, ConvNeXt_Base+CBAM had the highest sensitivity. Thus, it can be useful in providing assistance to professional ophthalmologists to obtain an accurate diagnosis. However, ConvNeXt_Base+CBAM and EfficientNet_V2-S+CBAM differ, EfficientNet_V2-S+CBAM has the best sensitivity and can assist other professionals, such as optometrists, family physicians, and non-ophthalmologists in screening patients who may have high myopia glaucoma.

Li et al. [36] and Cho et al. [55] included a C/D ratio of >0.7 to evaluate glaucoma and normal eyes. The AUCs of large-scale data sets were 0.986 and 0.975, both of which had a good performance. Previous studies have shown that a C/D ratio of >0.7 is characteristic of glaucoma [56], which may be one of the reasons why the model can improve glaucoma diagnosis. Among the inclusion conditions of this study, patients with high myopia and glaucoma were not restricted to a C/D ratio of >0.7. The C/D ratio of patients with high myopia is larger than that of people with normal eyes [14]; therefore, this is not specifically included in the condition. Shibata et al. [54] included only 55 patients with high myopia in a deep learning study. The images of high myopia with glaucoma can be diverse, and our research data can be more reliable when the number of images increases. Islam et al. [57] used four different deep-learning algorithms to diagnose glaucoma from cropped eye cup and fundus photographs. Among them, EfficientNet_b3 had the best test results, with an AUC score of 0.9512. In addition, satisfactory results were achieved using the U-net model for blood vessel segmentation on the full-eye color base map. However, although the cropped eye cup image can increase the training speed, it will sacrifice the possible sign impact of the images around the fundus. Therefore, the use of the whole fundus in our study can include more imaging parameters. Due to the black box effect in deep learning, we cannot fully understand the internal mechanism of the model. However, the accuracy of image diagnosis can be improved using the Grad-CAM technology. Further, an improved image quality can help improve diagnostic accuracy. In addition, we tried to verify the results of the ConvNeXt_Base+CBAM training model with the external data REFUGE test set. [58] The result had an AUC of 0.921, showing a certain degree of accuracy. We have demonstrated the use of deep learning to detect high myopic glaucoma, but there is still room for improvement. Kim et al. [37] used OCT data and combined demographic and ophthalmic characteristics, such as age, sex, axial length, and MD, to detect high myopic glaucoma. Therefore, a large amount of data resources and good results, with an AUC of 0.995, can be achieved. However, due to high equipment requirements, it is not easy to quickly screen and widely promote this method in clinical practice.

The study performed a Grad-CAM visual analysis of fundus photographs. The red areas in Figure 6a,c represent the pixels that contribute the most to the diagnostic results. Most pixels are located in the cup/disc area and neuroretinal rim and a few are located in the center of the macula, blood vessels, and other locations. This is consistent with the findings of ophthalmologists check when diagnosing glaucoma. In particular, the thermal imaging in parts of Figure 6a,b also focuses on the RNFL area. Additionally, it focuses on the highly myopic checkerboard-shaped fundus in Figure 6b,d.

In particular, this study excluded patients with pathological myopia, as there is a close relation between pathological myopia and glaucoma. Pathological myopia is defined as the presence of a myopic macula that is equal to or worse than diffuse chorioretinal atrophy. For local lesions [59], fundus features may be challenging to assess because the optic disc is severely tilted, deformed, and the optic nerve become thin. These features are similar to the pathological features of glaucoma, thereby making diagnosis more difficult. Future studies should consider including this patient population to completely evaluate the efficacy of deep-learning techniques in glaucoma diagnosis.

To the best of our knowledge, this is the first deep learning study to detect high myopic glaucoma from a large number of fundus photographs. Diagnosing glaucoma in ophthalmology requires multiple reference data and structural changes (intraocular pressure, OCT data, visual field data, fundus photographs, etc.), and diagnosis is only made after long-term observation. According to the myopia rate prediction report for 2050 [2], the incidence rate of high myopia among older adults will increase annually, and more people will develop age-related glaucoma. Consequently, glaucoma in highly myopic eyes will be challenging to diagnose and, currently, is a major public health issue. Our research used deep-learning methods and the fundus photography technology to address this difficulty. AI-assisted diagnosis can be applied in the field of ophthalmology in the future. For example, it can help non-ophthalmologists, such as optometrists and family physicians, make referrals or directly assist ophthalmologists in making accurate and rapid diagnoses. Overall, our study provides a feasible option for deep learning of fundus photographs to differentiate high myopia from glaucoma.

This study has some limitations. First, this study only included one ethnic group of Asians, and thus results should be interpreted with caution when applied to other ethnic groups. Second, we have not classified the degree of glaucoma. Different degrees of glaucoma are structurally different. Third, our data included no externally verified fundus photographs data; therefore, we cannot infer other image types, which reduce the generalization ability of the model. Fourth, there are insufficient OCT and visual field parameters in the high myopia group based on a medical record review. Although we have been extremely cautious in screening, these cases may still be misclassified. It is possible to increase sensitivity; however, glaucoma specialists should examine more eyes. Finally, patients with pathological myopia were excluded from the study. Hence, more clinicians will find it challenging to distinguish pathological myopia from glaucoma. Therefore, future studies should focus on this direction.

## 5. Conclusions

Glaucoma in individuals with high myopia was identified from their fundus photographs.

## Figures and Tables

**Figure 1 biomedicines-12-01394-f001:**
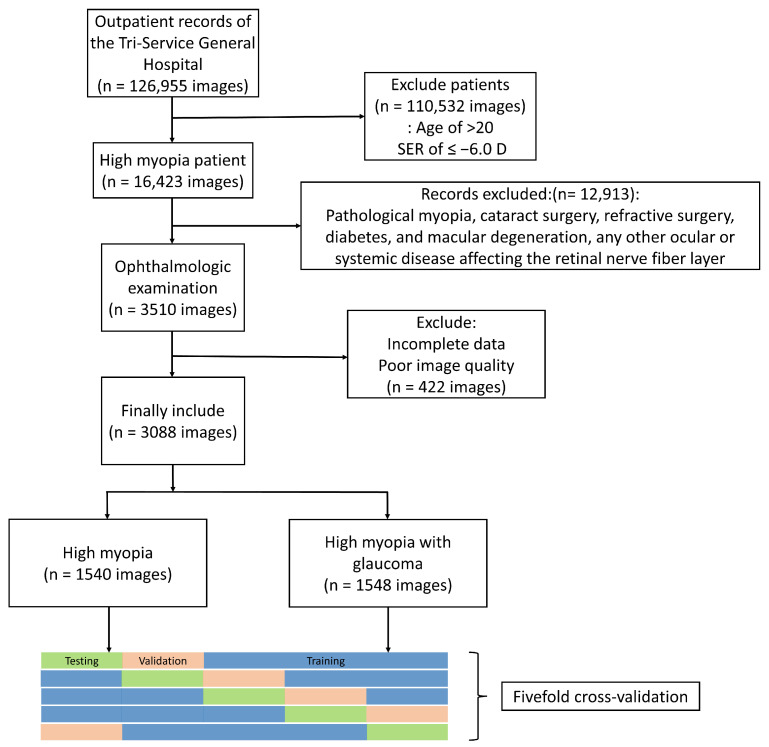
Flowchart of patient inclusion and exclusion.

**Figure 2 biomedicines-12-01394-f002:**
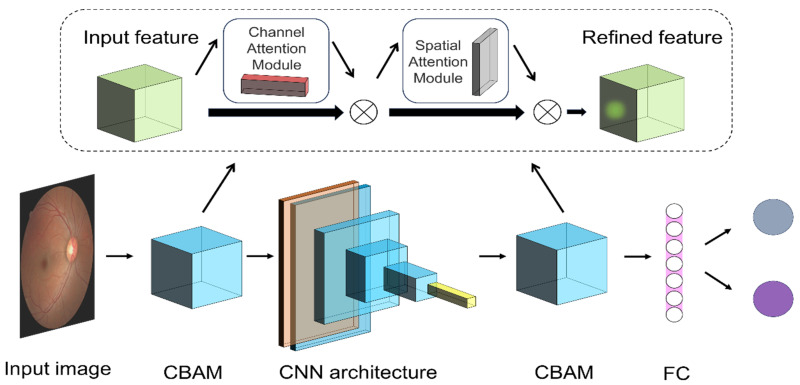
Convolutional neural network (CNN) and the convolutional block attention module (CBAM) attention mechanism. The image is input after preprocessing. Different CNN architectures were replaced.

**Figure 3 biomedicines-12-01394-f003:**
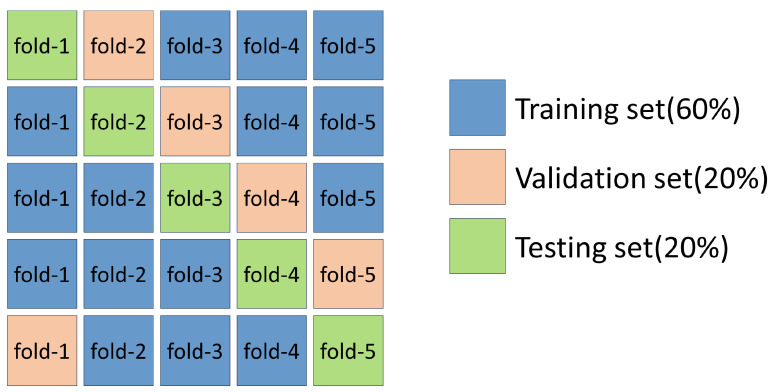
Fivefold cross-validation.

**Figure 4 biomedicines-12-01394-f004:**
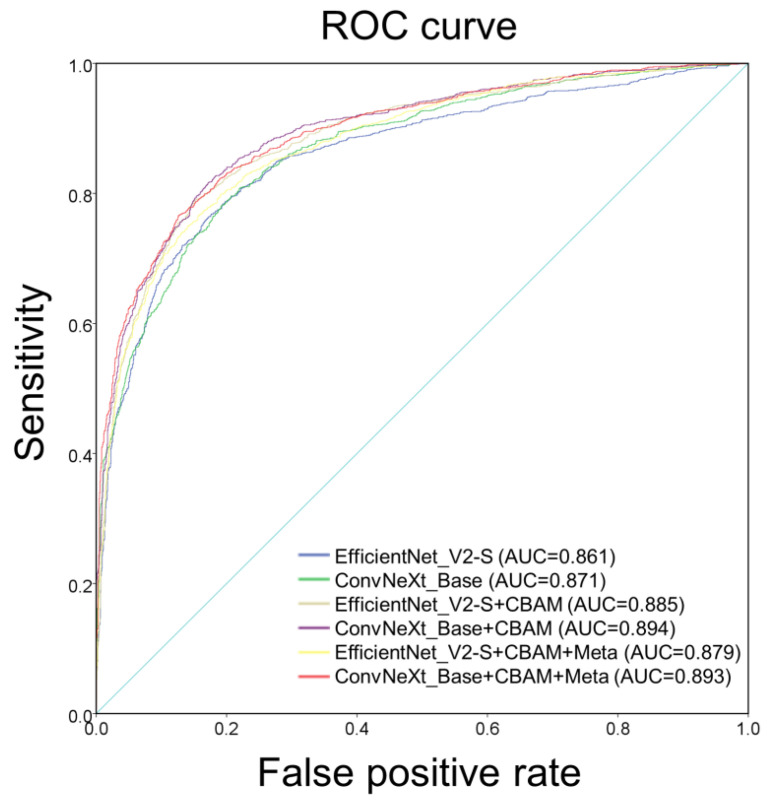
Operating characteristic curve of each classifier under different CNN architectures.

**Figure 5 biomedicines-12-01394-f005:**
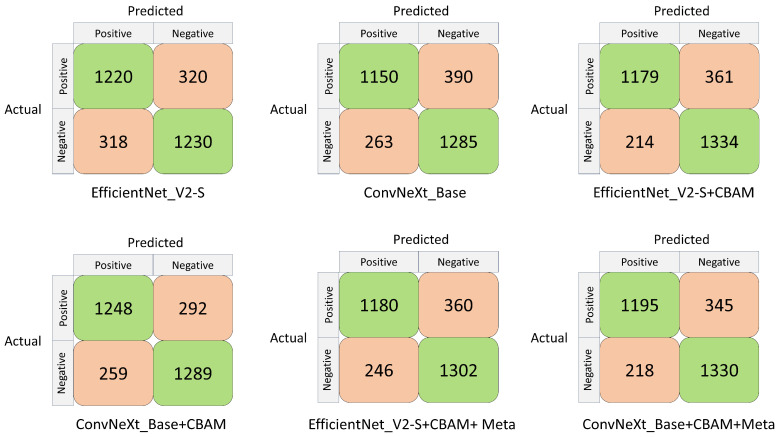
Confusion matrix of each classifier under different CNN architectures.

**Figure 6 biomedicines-12-01394-f006:**
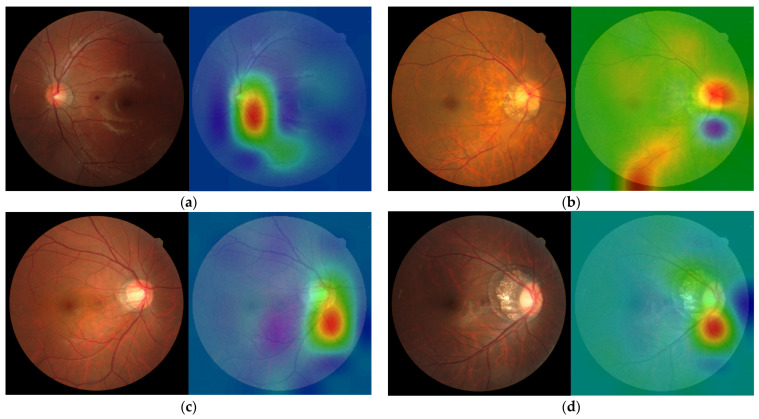
(**a**–**d**) Visual image heat maps. (**a**) Highly myopic left eye: diopter = −8.00 D, average RNFL = 105 µm, average C/D area ratio = 0.34, average MD = 0.78 dB, and normal GHT. (**b**) Highly myopic right eye: diopter = −8.50 D, average RNFL = 101 µm, average C/D area ratio = 0.41, average MD = 1.32 dB, and normal GHT. (**c**) High myopic glaucoma in the right eye: diopter = −10.25 D, average RNFL = 81 µm, average C/D area ratio = 0.56, average MD = −2.50 dB, and GHT outside the normal limits. (**d**) High myopic glaucoma in the right eye: diopter = −7.00 D, average RNFL = 76 µm, average C/D area ratio = 0.36, average MD = −2.21 dB, and GHT outside normal limits.

**Table 1 biomedicines-12-01394-t001:** Baseline demographics and clinical characteristics.

Variables	HM Glaucoma	HM	*p* Value
Patient	796	841	-
Eye (n)	1540	1548	-
Age (year)	47.6 ± 12.0	46.4 ± 14.2	**0.01**
Female (%)	48.77	59.04	**<0.001**
SE (diopter)	−8.83 ± 2.9	−8.73 ± 2.6	0.30
OCT parameters			
Average RNFL (um)	78.17 ± 15.2	-	-
Rim area (mm^2^)	0.9 ± 1.4	-	-
Disc area (mm^2^)	2.20 ± 2.9	-	-
Cup volume (mm^3^)	0.38 ± 0.38	-	-
Average C/D area ratio	0.71 ± 3.2	-	-
VF parameters			
Average MD (dB)	−6.15 ± 7.5	-	-
Average VFI (%)	84.88 ± 22.6	-	-

RNFL: retinal nerve fiber layer; VF: visual field; VFI: visual field index; C/D: cup/disc. All data are presented as mean ± standard deviation, unless otherwise stated. Statistically significant values are denoted in bold.

**Table 2 biomedicines-12-01394-t002:** Fivefold cross-validation data distribution.

	Train		Validation		Test	
	HMG	HM	HMG	HM	HMG	HM
Fold-1	924	930	308	309	308	309
Fold-2	924	930	308	309	308	309
Fold-3	924	930	308	309	308	309
Fold-4	924	927	308	312	308	309
Fold-5	924	927	308	309	308	312

HMG: high myopia glaucoma; HM: high myopia.

**Table 3 biomedicines-12-01394-t003:** Classification performance of convolutional neural networks.

Model	AUC	Accuracy	Sensitivity	Specificity	F1 Score
EfficientNet_V2-S	0.861	79.34%	79.22%	79.46%	79.27%
ConvNeXt_Base	0.870	78.85%	74.67%	83.01%	77.89%
EfficientNet_V2-S+CBAM	0.885	81.38%	76.56%	86.18%	80.40%
ConvNeXt_Base+CBAM	0.894	82.16%	81.04%	83.27%	81.92%
EfficientNet_V2-S+CBAM+Meta	0.879	80.38%	76.62%	84.11%	79.57%
ConvNeXt_Base+CBAM+Meta	0.893	81.77%	77.60%	85.92%	80.93%

Meta, gender, age, and diopter.

## Data Availability

The present data belong to and are stored at the Tri-Service General Hospital and cannot be shared without permission.

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
