# Peer review of "Deep Learning Evaluation of Glaucoma Detection Using Fundus Photographs in Highly Myopic Populations"

_biomedicines, 2024, doi:10.3390/biomedicines12071394_

Round 1

Reviewer 1 Report

Comments and Suggestions for Authors

The subject of the use of deep learning to diagnose glaucoma in high myopic patients can be of high interest, but the authors fail to present it in an appropriate detailed manner. It is not clear from the article what was the precise contribution of deep learning in diagnosing glaucoma in high myopic patients from fundus photographs. Therefore, I recommend a major review of this article with emphasis on the exact role of deep learning and a more detailed description of the modality in which it effectively helps clinicians in this difficult task. 

Author Response

RESPONSE TO REVIEWER COMMENTS

Comments 1:

The subject of the use of deep learning to diagnose glaucoma in high myopic patients can be of high interest, but the authors fail to present it in an appropriate detailed manner. It is not clear from the article what was the precise contribution of deep learning in diagnosing glaucoma in high myopic patients from fundus photographs. Therefore, I recommend a major review of this article with emphasis on the exact role of deep learning and a more detailed description of the modality in which it effectively helps clinicians in this difficult task. 

Response 1:

Thank you for your valuable feedback and constructive comments on our manuscript.  We appreciate your insights into the precise contribution of deep learning in diagnosing glaucoma in patients with high myopia from fundus photographs in more detail.  In response to your suggestions, we have made significant revisions to the introduction of the manuscript, specifically adding more detailed explanations in the second and fifth paragraphs. We will rewrite the introductory paragraph as follows:

"Axial elongation because of high myopia changes the retinal structure and function, including morphology similar to that seen in patients with glaucoma, such as disc size, shape, neuroretinal rim shape, and pallor. As high myopia worsens, the axial length of the eye elongates, and the optic disc stretches horizontally into an oval shape on fundus photography. Further, the condition may cause sagittal rotation of the optic disc, which is referred to as “tilted disc.” The normal optic disc shape and glaucoma are vertically elliptical. The cup-to-disc ratio in nonglaucoma high myopia is approximately 1.5 times higher than that in without myopia. However, previous research has shown that the optic disc size is not an important risk factor for the development of glaucoma. In myopia, the height between the neuroretinal edge and the base of the optic cup is reduced by flattening the optic cup. Simultaneously, the Bruch’s membrane opening expands, and the retinal edge becomes thinner. The neuroretinal edge loss in glaucoma progresses in inferotemporal, supratemporal, infranasal, and supranasal, respectively, as the disease progresses. High myopic glaucoma is more difficult to diagnose because of the abnormal shape of the optic retinal edge, which no longer conforms to the inferior–superior–nasal–temporal rule. High myopia and glaucoma cause thinning of the retinal nerve fiber layer. High myopia is mainly caused by the excessive elongation of the eyeball, and RNFL thinning is mainly observed in the superior and inferior temporal regions. Meanwhile, glaucoma is attributed to optic nerve damage caused by increased intraocular pressure. Further, the thickness of the RNFL and ganglion cell complex decreases significantly over time. According to the progression of visual field defects in myopia and glaucoma, non-myopic glaucoma usually manifests as Bjerrum area defects and nasal steps in the early stage. Further, central visual field defects develop over time. In the early stage of myopic glaucoma, central or paracentral darkening is more common. High myopia can be interpreted as false-positive glaucoma when diagnosing glaucoma. Fundus photography, Optical coherence tomography (OCT) is usually required to compare the results of long-term tracking changes with those of visual field examination. Therefore, diagnosing glaucoma in highly myopic eyes is difficult."

We rewrite the paragraph discussed as follows:

" To the best of our knowledge, this is the first deep learning study to detect high myopic glaucoma from a large number of fundus photographs. Diagnosing glaucoma in ophthalmology requires multiple reference data and structural changes (intraocular pressure, OCT data, visual field data, fundus photographs, etc.), and diagnosis is only made after long-term observation. According to the myopia rate prediction report for 2050, the incidence rate of high myopia among older adults will increase annually, and more people will develop age-related glaucoma. Consequently, glaucoma in highly myopic eyes will be challenging to diagnose and, currently, is a major public health issue. Our research used deep learning methods and the fundus photography technology to address this difficulty. AI-assisted diagnosis can be applied in the field of ophthalmology in the future. For example, it can help non-ophthalmologists, such as optometrists and family physicians, make referrals or directly assist ophthalmologists in making accurate and rapid diagnoses. Overall, our study provides a feasible option for deep learning of fundus photographs to differentiate high myopia from glaucoma. "

Please refer to page 2, lines 10-38 and page 12, lines 5-18.

         In these supplements, we emphasize the use of fundus changes in high myopia, as well as changes in glaucoma, both of which can create diagnostic dilemmas for clinicians. We elaborate on the benefits of using fundus photos for diagnosis using deep learning, such as avoiding data conversion problems between different machines, relatively fast processing time, price, and convenience. Furthermore, we highlight the core of our research The goal is to use deep learning to detect glaucoma from fundus photos of highly myopic individuals. Through deep learning, it helps ophthalmologists make diagnoses and promotes early referrals from optometrists and family doctors.

Reviewer 2 Report

Comments and Suggestions for Authors

This paper proposes a deep learning framework to identify glaucoma and normal eyes in groups with high myopia using fundus photographs. The authors retrospectively reviewed patients who visited Tri-Services General Hospital from November 1, 2018, to October 31, 2022. Patients with high myopia (spherical equivalent refraction of ≤ −6.0 D) were screened and divided into the high myopia group and the high myopia glaucoma group. We used two classification models with the convolutional block attention module (CBAM), an attention mechanism module that enhances the performance of convolutional neural networks (CNNs), to investigate glaucoma cases. The learning data of this experiment were evaluated through fivefold cross-validation. The images were categorized into training, validation, and test sets in a ratio of 6:2:2. Grad-CAM visual visualization improved the interpretability of the CNN results The subject matter is intriguing, however, the clarity of the paper's originality is uncertain. In addition, the related work section is missing which is a very important section.

The abstract: could you please highlight the novelty of the study?

Could you please change the word verification to validation?

Introduction

Please add a related work section discussing previous research that used deep learning for eye disease diagnosis

Please highlight novelty and contributions

Please add a paragraph describing the organization of the paper.

Methodology:

Please add samples of the dataset

Please mention how many patients were included in the study.

Justify the use of EfficientNet and ConvNeXt_Base

Experimental Results

Please dedicate a section defining performance measures.

The results section needs improvement. there are few results.

Please add sensitivity, F1-score, precision, and specificity metrics

Please also provide confusion metrics.

Please compare the results with other deep learning models

I cannot find a proper discussion on the results. Please add more discussions and dedicate a section for discussion.

Please add the limitations of this study

Author Response

RESPONSE TO REVIEWER COMMENTS

Comments 1:

Could you please highlight the novelty of the study?

Response 1:

Thank you for your feedback. The novelty of our study is emphasized in the introduction, and we rephrase that paragraph as follows:

" Previous deep learning studies have used non-highly myopic or OCT parameters for deep learning, or overly focused on the optic nerve head structure. Moreover, currently, no study has utilized fundus photos for evaluating high myopia glaucoma. The use of fundus photographs can prevent issues related to the inability to convert data between different machines such as OCT. The fundus photographs are also available easily and time-saving. This study aimed to use deep learning to detect glaucoma on the fundus photographs of individuals with high myopia. We believe that this method can assist ophthalmologists in achieving an accurate diagnosis or other professionals, such as optometrists, family physicians, and non-ophthalmologists, in making early referrals."

This change has been made on page 3, lines 14-22.

We rewrite the paragraph discussed as follows:

" To the best of our knowledge, this is the first deep learning study to detect high myopic glaucoma from a large number of fundus photographs. Diagnosing glaucoma in ophthalmology requires multiple reference data and structural changes (intraocular pressure, OCT data, visual field data, fundus photographs, etc.), and diagnosis is only made after long-term observation. According to the myopia rate prediction report for 2050, the incidence rate of high myopia among older adults will increase annually, and more people will develop age-related glaucoma. Consequently, glaucoma in highly myopic eyes will be challenging to diagnose and, currently, is a major public health issue. Our research used deep learning methods and the fundus photography technology to address this difficulty. AI-assisted diagnosis can be applied in the field of ophthalmology in the future. For example, it can help non-ophthalmologists, such as optometrists and family physicians, make referrals or directly assist ophthalmologists in making accurate and rapid diagnoses. Overall, our study provides a feasible option for deep learning of fundus photographs to differentiate high myopia from glaucoma. "

This change has been made on page 12, lines 5-18.

Comments 2:

Could you please change the word verification to validation?

Response 2:

Thanks for pointing this out. We have updated the manuscript to replace the term “validation” with “validation” to maintain consistency and accuracy of terminology. We rewrite the paragraph as follows: " The images were categorized into training, validation, and test sets in a ratio of 6:2:2. Grad-CAM visual visualization improved the interpretability of the CNN results."

This change was made on page 1, line 18.

Comments 3:

Please add a related work section discussing previous research that used deep learning for eye disease diagnosis

Response 3:

Thank you for your valuable feedback. We appreciate your suggestions and we have added a related work section to the manuscript to discuss previous studies utilizing deep learning for eye disease diagnosis. We rewrite the paragraph as follows: "In the study by Li et al, ophthalmologists classified 48,116 fundus photographs. Glaucoma was defined as a vertical cup-to-disc ratio of ≥ 0.7. In the Inception-v3 architecture, the AUC was 0.986, the accuracy was 92.9%, the sensitivity was 95.6%, and the specificity was 92.0%. In the study by Kim et al., 747 myopic healthy eyes and 1860 myopic glaucoma eyes were included, using macular vertical OCT to evaluate the glaucoma diagnostic ability of patients with high myopia. In the EfficientNet architecture, the external test dataset showed an AUC of 0.984 using macular vertical OCT scans and an AUC of 0.983 using individual data combinations. Asaoka et al. analyzed the visual field of the first diagnosis of evident glaucoma and the visual field of healthy people. Deep learning can be used to predict the development trend of glaucoma and achieve good outcomes (AUC: 92.6%). Regarding fundus photography medical imaging detection of glaucoma, previous studies focused more on the cutting of the optic disc and the structure of the optic nerve head, although they all have excellent performance. However, fundus photos with a viewing angle of 45º are not always used, and other signs in the eye map may be ignored.

Previous deep learning studies have used non-highly myopic or OCT parameters for deep learning, or overly focused on the optic nerve head structure. Moreover, currently, no study has utilized fundus photos for evaluating high myopia glaucoma. The use of fundus photographs can prevent issues related to the inability to convert data between different machines such as OCT. The fundus photographs are also available easily and time-saving. This study aimed to use deep learning to detect glaucoma on the fundus photographs of individuals with high myopia. We believe that this method can assist ophthalmologists in achieving an accurate diagnosis or other professionals, such as optometrists, family physicians, and non-ophthalmologists, in making early referrals."

Introduction (revised) See page 2, lines 53-54, and page 3, lines 1-22.

Comments 4:

Please highlight novelty and contributions

Response 4:

Thank you for your feedback. The novelty of our study is emphasized in the introduction, and we rephrase that paragraph as follows:

" Previous deep learning studies have used non-highly myopic or OCT parameters for deep learning, or overly focused on the optic nerve head structure. Moreover, currently, no study has utilized fundus photos for evaluating high myopia glaucoma. The use of fundus photographs can prevent issues related to the inability to convert data between different machines such as OCT. The fundus photographs are also available easily and time-saving. This study aimed to use deep learning to detect glaucoma on the fundus photographs of individuals with high myopia. We believe that this method can assist ophthalmologists in achieving an accurate diagnosis or other professionals, such as optometrists, family physicians, and non-ophthalmologists, in making early referrals."

This change has been made on page 3, lines 14-22.

We rewrite the paragraph discussed as follows:

" To the best of our knowledge, this is the first deep learning study to detect high myopic glaucoma from a large number of fundus photographs. Diagnosing glaucoma in ophthalmology requires multiple reference data and structural changes (intraocular pressure, OCT data, visual field data, fundus photographs, etc.), and diagnosis is only made after long-term observation. According to the myopia rate prediction report for 2050, the incidence rate of high myopia among older adults will increase annually, and more people will develop age-related glaucoma. Consequently, glaucoma in highly myopic eyes will be challenging to diagnose and, currently, is a major public health issue. Our research used deep learning methods and the fundus photography technology to address this difficulty. AI-assisted diagnosis can be applied in the field of ophthalmology in the future. For example, it can help non-ophthalmologists, such as optometrists and family physicians, make referrals or directly assist ophthalmologists in making accurate and rapid diagnoses. Overall, our study provides a feasible option for deep learning of fundus photographs to differentiate high myopia from glaucoma. "

This change has been made on page 12, lines 5-18.

The novelty and contributions of our study are as follows:

Novelty:

1.      This is the first study to utilize fundus photographs specifically for detecting glaucoma in highly myopic patients. Previous studies have primarily focused on non-high myopia cases or used OCT imaging.

Contributions:

1.      We developed a deep learning framework incorporating the convolutional block attention module (CBAM) to enhance CNN performance for glaucoma detection.

2.      The use of Grad-CAM visualization improved the interpretability of the CNN results, aiding in understanding how the model makes its predictions.

3.      Our study provides a valuable tool for assisting ophthalmologists and other healthcare professionals in diagnosing glaucoma in high myopia patients, potentially leading to earlier detection and treatment.

Comments 5:

Please add a paragraph describing the organization of the paper.

Response 5:

We have added a flowchart detailing the patient inclusion and exclusion process to provide clarity and guide readers through the manuscript.  Here is our description:

In the flowchart, we first introduce the initial inclusion of images (n = 126,955 images).  Next, we describe the criteria for patient inclusion (n = 16,423 images), exclusion by ophthalmology examination (n = 3,510 images), filtering for image quality (n = 3,088 images), and finally classification of images into the normal high myopia group. and high myopia glaucoma group, into the deep learning architecture. The reference Figure 1 is as follows:

Comments 6:

Please add samples of the dataset

Response 6:

In deep learning, the relationship between the number of training samples and the data utilized is crucial for model performance. Generally, a larger number of training samples enhances the model's ability to generalize, though it may also introduce the risk of overfitting. Increasing the training sample size typically improves model performance, as it allows the model to learn more comprehensive features from the data, leading to higher training and validation accuracy.

In our research, we employed data augmentation techniques to expand the existing dataset without acquiring new data. Specifically, we resized the images to 224 pixels (transforms.Resize(224)), randomly rotated them by up to 20 degrees (transforms.RandomRotation(20)), and applied random horizontal flips (transforms.RandomHorizontalFlip()). These operations enhance data diversity, thereby improving the model's generalization capability.

Effect of Sample Size on Model Accuracy:

In our preliminary study, we experimented with different sample sizes of 1000, 2000, 2500, and 3088 images for deep learning. The corresponding accuracies were 79.45%, 80.42%, 81.87%, and 82.16%. This indicates that beyond 2500 samples, the accuracy begins to stabilize. Given the retrospective nature of our study, we believe this sample size sufficiently demonstrates the efficacy of deep learning to a certain extent.

Comments 7:

Please mention how many patients were included in the study.

Response 7:

Thank you for your insightful comment. We rewrite the resulting paragraph as follows: " This study initially included 35,327 patients with 126,955 fundus photographs and 16,423 images with high myopia. After excluding patients who did not meet the conditions (n = 12,913) and those with poor image quality (n = 422), 1637 met the criteria. In total, there were 3088 fundus images, including those of 796 patients in the high myopia glaucoma group, with 1540 fundus images."

This change has been made on page 6, lines 36-38, and page 7, lines 1-2.

Comments 8:

Justify the use of EfficientNet and ConvNeXt_Base

Response 8:

Thank you for your thoughtful feedback.  We selected EfficientNet and ConvNeXt_Base because of their demonstrated performance and efficiency in a variety of computer vision tasks. EfficientNet is known for its ability to achieve high accuracy with fewer parameters and FLOPS, making it ideal for our research focused on optimizing resource usage without compromising accuracy.  ConvNeXt_Base, on the other hand, provides a modern architecture that effectively captures global and local features, thereby delivering superior performance on image classification benchmarks.  These choices are supported by the success of previous studies and compatibility with the requirements of our dataset.

For the description in the manuscript:

" A CNN retrieves image features for classification tasks through its deep structure, which greatly improves the accuracy and efficiency of disease diagnosis in medical image classification. Our study used two CNN architecture models: EfficientNet_V2-S, which was proposed by Tan and Le in 2021, and ConvNeXt_Base, which was introduced by Liu et al. in 2022. Additionally, we used transfer learning. Previous literature has revealed that pretrained parameters improve classification capabilities more quickly. " Please refer to lines 18-23 on page 4.

We rewrite the paragraph as follows:

" Although the incidence of high myopia is low, deep learning requires a large amount of data to improve performance of the model. Hence, pretraining was used, and it was combined with CBAM. Based on a previous research, we utilized CNN models with different architectures, such as EfficientNet_V2-S, ConvNeXt_Base, ResNet50, and ViT_B_16. The model with a higher accuracy in ImageNet Top-1 image classification was used as a basis for the initial training. Finally, EfficientNet_V2-S and ConvNeXt_Base, which have a better performance, were selected as the final model architecture. Among the models, ConvNeXt_Base+CBAM had the highest sensitivity. Thus, it can be useful in providing assistance to professional ophthalmologists to obtain an accurate diagnosis. However, ConvNeXt_Base+CBAM and EfficientNet_V2-S+CBAM differ, EfficientNet_V2-S+CBAM has the best sensitivity and can assist other professionals, such as optometrists, family physicians, and non-ophthalmologists in screening patients who may have high myopia glaucoma. "

This change has been made on page 11, lines 1-13.

Source of ImageNet Top-1 image classification: https://paperswithcode.com/sota/ image-classification-on-imagenet

Comments 9:

Please dedicate a section defining performance measures.

Response 9:

We've added a new section to define performance measures based on requirements. 

We rewrite the paragraph as follows:

" The metrics used to assess the model’s efficacy encompass the area under the curve (AUC) of the receiver operating characteristic (ROC), accuracy, sensitivity, specificity, and F1 score. The ROC curve depicts the balance between sensitivity and the complement of the false positive rate (1 - specificity). The AUC under the ROC curve was computed. An AUC value of 1.0 signifies flawless differentiation, whereas a value of 0.5 indicates discrimination equivalent to random chance. Accuracy is a metric that measures the proportion of correctly predicted instances among the total instances in a classification model. The F1 score is a metric that combines precision and sensitivity to evaluate the performance of a classification model, with values ranging from 0 to 1, where a higher score indicates a better model performance. The results of the fivefold cross-validation are averaged to assess generalizability. "

This can be found on page 6, lines 23-34.  We clearly define and explain all performance indicators used in our research, ensuring clarity and completeness.

Comments 10:

The results section needs improvement. there are few results.

Response 10:

We agree that the results section needs improvement. We have expanded the results section to include more comprehensive data and analysis. We rewrite the resulting paragraph as follows: " The OCT parameters of the high myopia glaucoma group were as follows: average RNFL, 78.17 ± 15.2 um; rim area, 0.9 ± 1.4 mm2; disc area, 2.20 ± 2.9 mm2; cup volume, 0.38 ± 0.38 mm3; and average C/D area ratio, 0.71 ± 3.2. The VF parameters of the high myopia glaucoma group were as follows: average MD, −6.15 ± 7.5 dB and average VFI, 84.88% ± 22.6% (Table 1). Table 2 shows the number of images in the training, validation, and test sets in the fivefold cross-validation."

" Table 3 presents the classification performance of the various learning networks. The two different model architectures, EfficientNet_V2-S and ConvNeXt_Base, demonstrated AUCs of 0.86 and 0.870, accuracy of 79.34% and 78.85%, sensitivities of 79.22% and 74.67%, and specificities of 79.46% and 83.01%, F1 score of 79.27% and 77.89%, respectively. The results of using CBAM in the model architecture (EfficientNet_V2-S+CBAM and ConvNeXt_Base+CBAM) indicated AUCs of 0.885 and 0.894, accuracy of 81.38% and 82.16%, sensitivities of 76.56% and 81.04%, and specificities of 86.18% and 83.27%, F1 score of 80.40% and 81.92%, respectively. The results after using CBAM in the model architecture and adding patient characteristics (EfficientNet_V2-S+CBAM+Meta and ConvNeXt_Base+CBAM+Meta) showed AUCs of 0.879 and 0.893, accuracy of 80.38% and 81.77%, sensitivities of 77.62% and 77.60%, and specificities of 84.11% and 85.95%, F1 score of 79.57% and 80.93%, respectively. Figure 4 illustrates the ROC curves and AUC values of various learning networks. "

"Figure 5 shows the confusion matrix of each classifier under different CNN architectures. The confusion matrices of EfficientNet_V2-S and ConvNeXt_Base were as follows: TP = 1220, 1150 images; FN = 320, 390 images; FP = 318, 263 images; and TN = 1230, 1285 images, respectively. The confusion matrices of EfficientNet_V2-S+CBAM and ConvNeXt_Base+CBAM were as follows: TP = 1179, 1248 images; FN = 361, 292 images; FP = 214, 259 images; and TN = 1334, 1289 images, respectively. The coefficient matrices of EfficientNet_V2-S+CBAM+Meta and ConvNeXt_Base+CBAM+Meta were as follows: TP = 1180, 1195 images; FN = 360, 345 images; FP = 246, 218 images; and TN = 1302, 1330 images, respectively. Figure 6 shows the visual image heat map under the ConvNeXt_Base+CBAM architecture. The upper row (a, b) shows high myopia and the lower row (c, d) exhibits high myopia glaucoma "

We have added Table 3. 

Table 3. Classification performance of convolutional neural networks

Model

AUC

Accuracy

Sensitivity

Specificity

F1 score

EfficientNet_V2-S

0.861

79.34%

79.22%

79.46%

79.27%

ConvNeXt_Base

0.870

78.85%

74.67%

83.01%

77.89%

EfficientNet_V2-S+CBAM

0.885

81.38%

76.56%

86.18%

80.40%

ConvNeXt_Base+CBAM

0.894

82.16%

81.04%

83.27%

81.92%

EfficientNet_V2-S+CBAM+ Meta

0.879

80.38%

76.62%

84.11%

79.57%

ConvNeXt_Base+CBAM+Meta

0.893

81.77%

77.60%

85.92%

80.93%

Meta, gender, age, and diopter

We have added Figure 5.

Figure 5. Confusion matrix of each classifier under different CNN architectures.

We have added Table 3. Found on page 7, lines 19-27, and page 8, lines 1-4. Added confusion matrix figure 5 and description, on page 8, lines 8-18.

Comments 11:

Please add sensitivity, F1-score, precision, and specificity metrics

Response 11:

Based on your suggestion, we included sensitivity, F1 score, precision, and specificity metrics in our performance evaluation. We rewrite the paragraph as follows:

" Table 3 presents the classification performance of the various learning networks. The two different model architectures, EfficientNet_V2-S and ConvNeXt_Base, demonstrated AUCs of 0.86 and 0.870, accuracy of 79.34% and 78.85%, sensitivities of 79.22% and 74.67%, and specificities of 79.46% and 83.01%, F1 score of 79.27% and 77.89%, respectively. The results of using CBAM in the model architecture (EfficientNet_V2-S+CBAM and ConvNeXt_Base+CBAM) indicated AUCs of 0.885 and 0.894, accuracy of 81.38% and 82.16%, sensitivities of 76.56% and 81.04%, and specificities of 86.18% and 83.27%, F1 score of 80.40% and 81.92%, respectively. The results after using CBAM in the model architecture and adding patient characteristics (EfficientNet_V2-S+CBAM+Meta and ConvNeXt_Base+CBAM+Meta) showed AUCs of 0.879 and 0.893, accuracy of 80.38% and 81.77%, sensitivities of 77.62% and 77.60%, and specificities of 84.11% and 85.95%, F1 score of 79.57% and 80.93%, respectively. Figure 4 illustrates the ROC curves and AUC values of various learning networks. "

We have added Table 3. 

Table 3. Classification performance of convolutional neural networks

Model

AUC

Accuracy

Sensitivity

Specificity

F1 score

EfficientNet_V2-S

0.861

79.34%

79.22%

79.46%

79.27%

ConvNeXt_Base

0.870

78.85%

74.67%

83.01%

77.89%

EfficientNet_V2-S+CBAM

0.885

81.38%

76.56%

86.18%

80.40%

ConvNeXt_Base+CBAM

0.894

82.16%

81.04%

83.27%

81.92%

EfficientNet_V2-S+CBAM+ Meta

0.879

80.38%

76.62%

84.11%

79.57%

ConvNeXt_Base+CBAM+Meta

0.893

81.77%

77.60%

85.92%

80.93%

Meta, gender, age, and diopter

Information on these indicators is presented on page 7, lines 19-27, and page 8, lines 1-4. We have added Table 3. Please refer to page 8, lines 5-6. We believe this addition will give us a deeper understanding of the model's performance.

Comments 12:

Please also provide confusion metrics.

Response 12:

We also provide confusion metrics upon your request.  The confusion metrics are shown in Figure 5 on page 9, lines 2-3. True positives, false positives, true negatives and false negatives are clearly outlined in Figure and the results of the confusion matrix can be found on page 8, lines 8-18.

Figure 5. Confusion matrix of each classifier under different CNN architectures.

Comments 13:

Please compare the results with other deep learning models. I cannot find a proper discussion on the results. Please add more discussions and dedicate a section for discussion.

Response 13:

Thank you for your valuable feedback.  We apologize for the oversight and agree that a comparative analysis with other deep learning models will give us a more comprehensive understanding of the results.  We now compare our results in detail with those of several other state-of-the-art deep learning models.

" Li et al. and Cho et al. included a C/D ratio of >0.7 to evaluate glaucoma and normal eyes. The AUCs of large-scale data sets were 0.986 and 0.975, both of which had a good performance. Previous studies have shown that a C/D ratio of >0.7 is characteristic of glaucoma, which may be one of the reasons why the model can improve glaucoma diagnosis. Among the inclusion conditions of this study, patients with high myopia and glaucoma were not restricted to a C/D ratio of >0.7. The C/D ratio of patients with high myopia is larger than that of people with normal eyes; therefore, this is not specifically included in the condition. Shibata et al. included only 55 patients with high myopia in a deep learning study. The images of high myopia with glaucoma can be diverse, and our research data can be more reliable when the number of images increases. Islam et al. used four different deep learning algorithms to diagnose glaucoma from cropped eye cup and fundus photographs. Among them, EfficientNet_b3 had the best test results, with an AUC score of 0.9512. In addition, satisfactory results were achieved using the U-net model for blood vessel segmentation on the full-eye color base map. However, although the cropped eye cup image can increase the training speed, it will sacrifice the possible sign impact of the images around the fundus. Therefore, the use of the whole fundus in our study can include more imaging parameters. Due to the black box effect in deep learning, we cannot fully understand the internal mechanism of the model. However, the accuracy of image diagnosis can be improved using the Grad-CAM technology. Further, an improved image quality can help improve diagnostic accuracy. In addition, we tried to verify the results of the ConvNeXt_Base+CBAM training model with the external data REFUGE test set. The result had an AUC of 0.921, showing a certain degree of accuracy. We have demonstrated the use of deep learning to detect high myopic glaucoma, but there is still room for improvement. Kim et al. used OCT data and combined demographic and ophthalmic characteristics, such as age, sex, axial length, and MD, to detect high myopic glaucoma. Therefore, a large amount of data resources and good results, with an AUC of 0.995, can be achieved. However, due to high equipment requirements, it is not easy to quickly screen and widely promote this method in clinical practice."

This new discussion is on page 11, lines 14-42.  We believe these additions enhance the clarity and significance of our study.

Comments 14:

Please add the limitations of this study

Response 14:

We appreciate your suggestion to discuss the limitations of our study. Acknowledging limitations is crucial for the integrity of scientific research. We have added a paragraph on page 12, lines 19-30, detailing the following limitations:

" This study has some limitations. First, this study only included one ethnic group of Asians, and thus results should be interpreted with caution when applied to other ethnic groups. Second, we have not classified the degree of glaucoma. Different degrees of glaucoma are structurally different. Third, our data included no externally verified fundus photographs data; therefore, we cannot infer other image types, which reduce the generalization ability of the model. Fourth, there are insufficient OCT and visual field parameters in the high myopia group based on a medical record review. Although we have been extremely cautious in screening, these cases may still be misclassified. It is possible to increase sensitivity; however, glaucoma specialists should examine more eyes. Finally, patients with pathological myopia were excluded from the study. Hence, more clinicians will find it challenging to distinguish pathological myopia from glaucoma. Therefore, future studies should focus on this direction. "

Reviewer 3 Report

Comments and Suggestions for Authors

Many thanks to the editor for an opportunity to review the paper. The manuscript is well-written and describe interesting strategy for an important question. However some issues are raised by the methodology of segregation of cases with glaucoma.

Please consider that disc size in not a feature of glaucoma, it rather complicates diagnosis but does not associate with glaucoma diagnosis or severity.

“To compare the progression of the disease in different periods, our inclusion criteria for glaucoma patients are” this phrase in not clear. Why you need “To compare the progression of the disease in different periods”. It looks like inclusion criteria but no VF data presented here. Why?

“(1) changes in glaucomatous optic nerve or nerve fiber layer defects;” how this was established?

“; (2) there is a visual field progression defect corresponding to glaucoma during automated visual field examination;” in all 1627 subjects?

“(3) receiving antiglaucoma treatment (such as the following eye drops: beta blockers, alpha-2 adrenergic agonists, prostaglandin analogs, and carbonic anhydrase inhibitors)” cannot be consider as a factor indicating presence of glaucoma.

Why no OCT data provided for HM in table 2?

Please provide OCT and VF data confirming presence and absence of glaucoma for the cases presented in figure 4.

Author Response

RESPONSE TO REVIEWER COMMENTS

Comments 1:

Many thanks to the editor for an opportunity to review the paper. The manuscript is well-written and describe interesting strategy for an important question. However some issues are raised by the methodology of segregation of cases with glaucoma.

Response 1:

Thank you for your positive feedback on our manuscript.  We acknowledge concerns about the approach to glaucoma case isolation. We rewrite the paragraph as follows: " All patients with glaucoma were diagnosed after long-term follow-up by professional ophthalmologists. The inclusion criteria for patients with glaucoma were as follows: First, changes that indicate glaucomatous optic nerve or nerve fiber layer defects, such as increased narrowing of the neuroretinal rim (change in a sector of the neuroretinal rim from narrow to complete loss or from a homogeneous neuroretinal rim to a narrow sector) and a significant expansion of a retinal nerve fiber layer defect on OCT. Second, automated visual field testing showed glaucomatous visual field progression defined using the Anderson-Patella criteria, i.e., the Glaucoma Hemifield Test (GHT) result was outside normal limits. There was a cluster of three or more non-edge points in the typical location of the glaucoma, all of which were depressed on the pattern deviation plot at a p value of <5%, with at least one of these points depressed at a p value of <1%. In addition, the corrected pattern standard deviation was abnormal at a p value of <5%. "

We have revised the manuscript to provide a more detailed explanation of our isolation criteria and methods on page 3, lines 51-52, and page 4, lines 1-10.  The revised text now clarifies the steps and criteria used to ensure accurate case isolation.

Comments 2:

Please consider that disc size in not a feature of glaucoma, it rather complicates diagnosis but does not associate with glaucoma diagnosis or severity.

Response 2:

We appreciate this important clarification.  We have revised the manuscript to reflect that disc size, while complicating diagnosis, is not directly related to glaucoma diagnosis or severity. We rewrite the paragraph as follows: " As high myopia worsens, the axial length of the eye elongates, and the optic disc stretches horizontally into an oval shape on fundus photography. Further, the condition may cause sagittal rotation of the optic disc, which is referred to as “tilted disc.” The normal optic disc shape and glaucoma are vertically elliptical. The cup-to-disc ratio in nonglaucoma high myopia is approximately 1.5 times higher than that in without myopia. However, previous research has shown that the optic disc size is not an important risk factor for the development of glaucoma. "

This revision can be found on page 2, lines 12-18.  The updated text now emphasizes that optic disc size is considered in the context of diagnostic complexity rather than as a direct feature of glaucoma.

Comments 3:

“To compare the progression of the disease in different periods, our inclusion criteria for glaucoma patients are” this phrase in not clear. Why you need “To compare the progression of the disease in different periods”. It looks like inclusion criteria but no VF data presented here. Why?

Response 3:

Thank you for pointing out the ambiguity in our wording.  We have rephrased the sentences on page 3, lines 51-52, and page 4, lines 1-10, to clearly express our intent. The revised text is now as follows: " All patients with glaucoma were diagnosed after long-term follow-up by professional ophthalmologists. The inclusion criteria for patients with glaucoma were as follows: First, changes that indicate glaucomatous optic nerve or nerve fiber layer defects, such as increased narrowing of the neuroretinal rim (change in a sector of the neuroretinal rim from narrow to complete loss or from a homogeneous neuroretinal rim to a narrow sector) and a significant expansion of a retinal nerve fiber layer defect on OCT. Second, automated visual field testing showed glaucomatous visual field progression defined using the Anderson-Patella criteria, i.e., the Glaucoma Hemifield Test (GHT) result was outside normal limits. There was a cluster of three or more non-edge points in the typical location of the glaucoma, all of which were depressed on the pattern deviation plot at a p value of <5%, with at least one of these points depressed at a p value of <1%. In addition, the corrected pattern standard deviation was abnormal at a p value of <5%. "

The visual field data are now included in the revised manuscript, lines 5-10 on page 7. We rewrite the paragraph as follows: " The OCT parameters of the high myopia glaucoma group were as follows: average RNFL, 78.17 ± 15.2 um; rim area, 0.9 ± 1.4 mm2; disc area, 2.20 ± 2.9 mm2; cup volume, 0.38 ± 0.38 mm3; and average C/D area ratio, 0.71 ± 3.2. The VF parameters of the high myopia glaucoma group were as follows: average MD, −6.15 ± 7.5 dB and average VFI, 84.88% ± 22.6% (Table 1). Table 2 shows the number of images in the training, validation, and test sets in the fivefold cross-validation."

Because this is a retrospective study, the medical records of the high myopia group do not have complete visual field data or complete OCT data. Among patients in the high myopia group, who are considered to be low risk after examination by a glaucoma physician, no visual field examination will be arranged. In this group, we are indeed not sure whether they have glaucoma. This is our limitation and we cannot rule out possible misclassifications. Among the restrictions, we rewrote a section as follows: " Fourth, there are insufficient OCT and visual field parameters in the high myopia group based on a medical record review. Although we have been extremely cautious in screening, these cases may still be misclassified. It is possible to increase sensitivity; however, glaucoma specialists should examine more eyes."

This change has been made on page 12, lines 24-27.

Comments 4:

“(1) changes in glaucomatous optic nerve or nerve fiber layer defects;” how this was established?

Response 4:

As in the reply above, we explain in detail how changes in the optic nerve or nerve fiber layer defects in glaucoma are determined. We have rephrased the sentences on page 3, lines 51-52, and page 4, lines 1-10, to clearly express our intent.  The revised text is now as follows: " All patients with glaucoma were diagnosed after long-term follow-up by professional ophthalmologists. The inclusion criteria for patients with glaucoma were as follows: First, changes that indicate glaucomatous optic nerve or nerve fiber layer defects, such as increased narrowing of the neuroretinal rim (change in a sector of the neuroretinal rim from narrow to complete loss or from a homogeneous neuroretinal rim to a narrow sector) and a significant expansion of a retinal nerve fiber layer defect on OCT. Second, automated visual field testing showed glaucomatous visual field progression defined using the Anderson-Patella criteria, i.e., the Glaucoma Hemifield Test (GHT) result was outside normal limits. There was a cluster of three or more non-edge points in the typical location of the glaucoma, all of which were depressed on the pattern deviation plot at a p value of <5%, with at least one of these points depressed at a p value of <1%. In addition, the corrected pattern standard deviation was abnormal at a p value of <5%. "

Comments 5:

“(2) there is a visual field progression defect corresponding to glaucoma during automated visual field examination” in all 1627 subjects? 

Response 5:

We apologize for the oversight.  We have clarified that all 796 glaucoma subjects had confirmed visual field progression defects during automated perimetry.

" The OCT parameters of the high myopia glaucoma group were as follows: average RNFL, 78.17 ± 15.2 um; rim area, 0.9 ± 1.4 mm2; disc area, 2.20 ± 2.9 mm2; cup volume, 0.38 ± 0.38 mm3; and average C/D area ratio, 0.71 ± 3.2. The VF parameters of the high myopia glaucoma group were as follows: average MD, −6.15 ± 7.5 dB and average VFI, 84.88% ± 22.6% (Table 1). Table 2 shows the number of images in the training, validation, and test sets in the fivefold cross-validation."

This description can be found on page 7, lines 5-10. 

Since this is a retrospective study, the medical records of the highly myopic group do not have complete visual field data or complete OCT data. Among the highly myopic group, patients who are considered to be at low risk after examination by a glaucoma physician will not be scheduled for visual field examination. Under Taiwan's health insurance system, hospital visual field examinations and OCT examinations cannot be performed at will.

This is mainly due to the following reasons:

First, a visual field examination is a must at the time of initial diagnosis, as early identification of the condition is critical to the development of subsequent treatment options. But when the condition is stable, annual checkups are usually only needed. This can not only effectively monitor the condition, but also avoid unnecessary repeated examinations, reduce patient trouble and waste of medical resources.

Secondly, although OCT examination is also an item covered by health insurance, its arrangement needs to be decided by the doctor based on the specific situation. Doctors will determine the necessity and frequency of examinations based on the patient's actual condition to ensure that each examination can provide valuable diagnostic information, rather than examinations just for the sake of examination.

The arrangement of these examinations is based on medical professional judgment and the actual needs of the patient, with the aim of providing the most effective diagnosis and treatment options.

Reference health insurance information:

https://sheethub.com/data.gov.tw/%E9%86%AB%E7%99%82%E6%9C%8D%E5%8B%99%E7%B5%A6%E4%BB%98 %E9%A0%85%E7%9B%AE?page=29

In the highly myopic group, we really don't know for sure whether they have glaucoma. This is a limitation of ours and we cannot rule out possible misclassification. Among them, we rewrote a paragraph as follows: " Fourth, there are insufficient OCT and visual field parameters in the high myopia group based on a medical record review. Although we have been extremely cautious in screening, these cases may still be misclassified. It is possible to increase sensitivity; however, glaucoma specialists should examine more eyes. "

This change has been made on page 12, lines 24-27.

Comments 6:

(3) receiving antiglaucoma treatment (such as the following eye drops: beta blockers, alpha-2 adrenergic agonists, prostaglandin analogs, and carbonic anhydrase inhibitors)” cannot be consider as a factor indicating presence of glaucoma.

Response 6:

We appreciate this important correction. We have revised the manuscript to remove the consideration of receiving antiglaucoma treatment as an indicator of glaucoma presence.

We rewrite the paragraph as follows: " All patients with glaucoma are receiving antiglaucoma treatment, such as the following eye drops: beta blockers, alpha-2 adrenergic agonists, prostaglandin analogs, and carbonic anhydrase inhibitors. "

This revision is reflected on page 4, lines 10-13. The text now accurately distinguishes between treatment and diagnostic criteria.

Comments 7:

Why no OCT data provided for HM in table 2?

Response 7:

We sincerely apologize, we have clarified and responded uniformly in Response 5.

Comments 8:

Please provide OCT and VF data confirming presence and absence of glaucoma for the cases presented in figure 4.

Response 8:

We apologize for the incomplete data presentation. We have now included the OCT and VF data confirming the presence and absence of glaucoma for the cases presented in Figure 6. (Originally Figure 4)  We rewrite it as follows: " (a), (b), (c), and (d) Visual image heat maps. (a) Highly myopic left eye: diopter = −8.00 D, average RNFL = 105 um, average C/D area ratio = 0.34, average MD = 0.78 dB, and normal GHT. (b) Highly myopic right eye: diopter = −8.50 D, average RNFL = 101 um, average C/D area ratio = 0.41, average MD = 1.32 dB, and normal GHT. (c) High myopic glaucoma in the right eye: diopter = −10.25 D, average RNFL = 81 um, average C/D area ratio = 0.56, average MD = −2.50 dB, and GHT outside the normal limits. (d) High myopic glaucoma in the right eye: diopter = −7.00 D, average RNFL = 76 um, average C/D area ratio = 0.36, average MD = −2.21 dB, and GHT outside normal limits. "

This information can be found on page 10, lines 5- 10, providing a complete and transparent dataset for the cases discussed.

Reviewer 4 Report

Comments and Suggestions for Authors

In the abstract it is not explained sufficiently that the configuration ConvNeXt_Base+CBAM was the one with the highest AUC, the reference to "various learning networks" confuses the phrase.

In the Introduction: "Many symptoms can confuse the diagnosis...especially in the morphology" but the morphology is a clinical sign, not a symptom.

"it is difficult to make a diagnosis of glaucoma based only on the size of the optic disc" - actually, it is impossible to do that, the phrase was not formulated well

"focus more on the cutting of the optic disk, the size of the optic disk, and the optic nerve head" - do you mean cupping of the optic disk? And the disk or the optic nerve head are not the same thing?

In Materials and methods:

"This study exclude patients with pathological myopia" -I believe that it a big issue since the diagnosis of glaucoma in a pathological myopia would be much more difficult, so the help of the deep learning technology would be more appealing to the clinicians. I believe you should  acknowledge that in the abstract!

"To compare the progression of the disease in different periods" ...this has nothing to do with the inclusion criteria, please rephrase.

In Results:

"this study met the criteria" - this affirmation has no sense, the patients included are the ones that met the criteria

In Discussion:

"EfficientNet_V2-S+CBAM+Meta and Con-vNeXt_Base+CBAM+Meta added with more patient characteristics" - do you mean augmented with more characteristics?

"more people may be troubled by glaucoma along with age-related glaucoma" - difficult to understand the meaning, please rephrase

"OCT and visual field parameters were inadequate for high myopia," - it would be more accurate to say that the diagnostic value of OCT and of the visual field are reduced by the presence of high myopia

"fundus photography is implemented in regions" - please rephrase

Finally, I would ask if there was any possibility to increase the sensitivity, with the risk of having more eyes checked by a glaucoma specialist (but reducing the risk that this screening technique would miss a glaucomatous eye).

Comments on the Quality of English Language

There are several phrases that are difficult to comprehend, some of which I have already signaled.  I suggest a complete English language editing.

Author Response

RESPONSE TO REVIEWER COMMENTS

Comments 1:

In the abstract it is not explained sufficiently that the configuration ConvNeXt_Base+CBAM was the one with the highest AUC, the reference to "various learning networks" confuses the phrase.

Response 1:

Thank you for your valuable feedback. We appreciate your observation regarding the abstract.

We rewrite the paragraph as follows: " Based on a fivefold cross-validation assessment, the ConvNeXt_Base+CBAM architecture had the best performance, with an AUC of 0.894, accuracy of 82.16%, sensitivity of 81.04%, specificity of 83.27%, and F1 score of 81.92%."

On page 1, line 25-27, we will revise the abstract to clearly state that the configuration ConvNeXt_Base+CBAM achieved the highest AUC. The phrase "various learning networks" will be clarified to avoid confusion.

Comments 2:

In the Introduction: "Many symptoms can confuse the diagnosis...especially in the morphology" but the morphology is a clinical sign, not a symptom.

Response 2:

Thank you for pointing out the inaccuracy. In the Introduction, on page 2, lines 12-18, we will revise the phrase to: "As high myopia worsens, the axial length of the eye elongates, and the optic disc stretches horizontally into an oval shape on fundus photography. Further, the condition may cause sagittal rotation of the optic disc, which is referred to as “tilted disc.” The normal optic disc shape and glaucoma are vertically elliptical. The cup-to-disc ratio in nonglaucoma high myopia is approximately 1.5 times higher than that in without myopia. However, previous research has shown that the optic disc size is not an important risk factor for the development of glaucoma. "

Comments 3:

"it is difficult to make a diagnosis of glaucoma based only on the size of the optic disc" - actually, it is impossible to do that, the phrase was not formulated well

Response 3:

We appreciate your comments. you are right. On page 2, lines 15-18, we reword the morphological sign difference and emphasize that " The normal optic disc shape and glaucoma are vertically elliptical. The cup-to-disc ratio in nonglaucoma high myopia is approximately 1.5 times higher than that in without myopia. However, previous research has shown that the optic disc size is not an important risk factor for the development of glaucoma."

Comments 4:

"focus more on the cutting of the optic disk, the size of the optic disk, and the optic nerve head" - do you mean cupping of the optic disk? And the disk or the optic nerve head are not the same thing?

Response 4:

Thank you for your comment. We apologize for the confusion. On page 3, lines 9-11, we will clarify the statement to: "Regarding fundus photography medical imaging detection of glaucoma, previous studies focused more on the cutting of the optic disc and the structure of the optic nerve head, although they all have excellent performance. "

Comments 5:

"This study exclude patients with pathological myopia" -I believe that it a big issue since the diagnosis of glaucoma in a pathological myopia would be much more difficult, so the help of the deep learning technology would be more appealing to the clinicians. I believe you should  acknowledge that in the abstract!

Response 5:

Thank you for your valuable feedback. We have rewritten a paragraph in the abstract as follows: " Patients who visited Tri-Services General Hospital from November 1, 2018, to October 31, 2022, were retrospectively reviewed. Patients with high myopia (spherical equivalent refraction of ≤ −6.0 D) were included in the current analysis. Meanwhile, patients with pathological myopia were excluded. The participants were then divided into the high myopia group and high myopia glaucoma group." This change has been made on page 1, lines 10-14.

We acknowledge the importance of including patients with pathological myopia, as the diagnosis of glaucoma in such cases is indeed more challenging, and deep learning technology could significantly aid clinicians in this regard. In response to your suggestion, we have added the following statement to acknowledge this point:

"In particular, this study excluded patients with pathological myopia, as there is a close relation between pathological myopia and glaucoma. Pathological myopia is defined as the presence of a myopic macula that is equal to or worse than diffuse chorioretinal atrophy. For local lesions, fundus features may be challenging to assess because the optic disc is severely tilted, deformed, and the optic nerve become thin. These features are similar to the pathological features of glaucoma, thereby making diagnosis more difficult. Future studies should consider including this patient population to completely evaluate the efficacy of deep learning techniques in glaucoma diagnosis"

This addition can be found on page 11, lines 51-54, and page 12, lines 1-4 of the revised manuscript.

Comments 6:

"To compare the progression of the disease in different periods" ...this has nothing to do with the inclusion criteria, please rephrase.

Response 6:

Thank you for your suggestion. We will rephrase the sentence on page 3, lines 51-52 to: " All patients with glaucoma were diagnosed after long-term follow-up by professional ophthalmologists. The inclusion criteria for patients with glaucoma were as follows:..."

Comments 7:

"this study met the criteria" - this affirmation has no sense, the patients included are the ones that met the criteria

Response 7:

We appreciate your comments. On lines 36-38 on page 6 and lines 1-2 on page 7, we will change the statement to: " This study initially included 35,327 patients with 126,955 fundus photographs and 16,423 images with high myopia. After excluding patients who did not meet the conditions (n = 12,913) and those with poor image quality (n = 422), 1637 met the criteria. In total, there were 3088 fundus images, including those of 796 patients in the high myopia glaucoma group, with 1540 fundus images. "

Comments 8:

"EfficientNet_V2-S+CBAM+Meta and Con-vNeXt_Base+CBAM+Meta added with more patient characteristics" - do you mean augmented with more characteristics?「EfficientNet_V2-S+CBAM+Meta 和 Con-vNeXt_Base+CBAM+Meta

Response 8:

Thank you for the clarification.  On page 10, lines 27-29, we describe " EfficientNet_V2-S+CBAM+Meta and ConvNeXt_Base+CBAM+Meta added with more patient characteristics, including gender, age, and diopter, demonstrated AUC values of 0.879 and 0.893, respectively."

Comments 9:

"more people may be troubled by glaucoma along with age-related glaucoma" - difficult to understand the meaning, please rephrase

Response 9:

Thank you for your feedback. We will rephrase the sentence on page 12, lines 9-11 to: " According to the myopia rate prediction report for 2050, the incidence rate of high myopia among older adults will increase annually, and more people will develop age-related glaucoma."

Comments 10:

"OCT and visual field parameters were inadequate for high myopia," - it would be more accurate to say that the diagnostic value of OCT and of the visual field are reduced by the presence of high myopia

Response 10:

We appreciate your suggestion. On page 12, lines 6-9, we will revise the sentence to: " Diagnosing glaucoma in ophthalmology requires multiple reference data and structural changes (intraocular pressure, OCT data, visual field data, fundus photographs, etc.), and diagnosis is only made after long-term observation."

Comments 11:

"fundus photography is implemented in regions" - please rephrase

Response 11:

Thank you for your comment. We rewrite the paragraph as follows: " Our research used deep learning methods and the fundus photography technology to address this difficulty. AI-assisted diagnosis can be applied in the field of ophthalmology in the future. For example, it can help non-ophthalmologists, such as optometrists and family physicians, make referrals or directly assist ophthalmologists in making accurate and rapid diagnoses. "

This change has been made on page 12, lines 13-17.

Comments 12:

Finally, I would ask if there was any possibility to increase the sensitivity, with the risk of having more eyes checked by a glaucoma specialist (but reducing the risk that this screening technique would miss a glaucomatous eye).

Response 12:

This is a good suggestion, so we'll add your suggestion to the Limits paragraph of our Discussion. We rewrite the paragraph as follows: " Fourth, there are insufficient OCT and visual field parameters in the high myopia group based on a medical record review. Although we have been extremely cautious in screening, these cases may still be misclassified. It is possible to increase sensitivity; however, glaucoma specialists should examine more eyes."

This change has been made on page 12, lines 24-27.

It is indeed possible to improve the sensitivity of screening technology when every subject is examined by a glaucoma specialist and performs OCT and VF examinations. However, not every patient in our study had such an examination completely performed on their eyes. However, we carefully selected each patient with high myopia to avoid misclassification. These patients were diagnosed as low-risk glaucoma after examination by an ophthalmologist.

4. Response to Comments on the Quality of English Language

Point 1:

There are several phrases that are difficult to comprehend, some of which I have already signaled.  I suggest a complete English language editing.

Response 1:

Thank you for your valuable feedback regarding the quality of the English language in our manuscript. We deeply appreciate your thorough review and the time you took to highlight specific areas that require improvement.We have carefully reviewed and revised the entire manuscript to enhance clarity and readability. We have made specific changes to address the above issues.

Round 2

Reviewer 1 Report

Comments and Suggestions for Authors

The paper can be published in Biomedicines.

Author Response

Thank you very much for your positive feedback and recommendation. We are grateful for your thorough review and are pleased to hear that our work meets the standards of the journal.

Reviewer 2 Report

Comments and Suggestions for Authors

Thank you for addressing most of my comments. However, there are few comments that have not been properly addressed.

Please summarize the novelty and contributions in form of points and add them to the end of the introduction section.

please add a paragraph by the end of the introduction that describes the structure and organization of the sections of the manuscript.

please add some images to different classes of the dataset in the dataset description section

Author Response

RESPONSE TO REVIEWER COMMENTS

Comments 1:

Please summarize the novelty and contributions in form of points and add them to the end of the introduction section.

Response 1:

Thank you for your feedback. The novelty of our study is emphasized in the introduction, and we rephrase that paragraph as follows:

" Previous deep learning studies have used non-highly myopic or OCT parameters for deep learning, or overly focused on the optic nerve head structure. Moreover, currently, no study has utilized fundus photos for evaluating high myopia glaucoma. Herein, several novel issues were explored. First, this is the study to specifically use fundus photographs for detecting glaucoma in patients with high myopia. Second, we develop a deep learning framework incorporating convolutional block attention modules (CBAM) to enhance the performance of CNN glaucoma detection. Finally, the use of Grad-CAM visualization im-proves the interpretability of CNN results and helps comprehend how the model makes predictions. The use of fundus photographs can prevent issues related to the inability to convert data between different machines such as OCT. The fundus photographs are also available easily and time-saving. This study aimed to use deep learning to detect glaucoma on the fundus photographs of individuals with high myopia. We believe that this method can assist ophthalmologists in achieving an accurate diagnosis or other professionals, such as optometrists, family physicians, and non-ophthalmologists, in making early referrals."

This change has been made on page 3, lines 14-28.

Comments 2:

please add a paragraph by the end of the introduction that describes the structure and organization of the sections of the manuscript.

Response 2:

Thank you for your detailed review of the manuscript. Your suggestion to add a paragraph at the end of the introduction describing the structure and organization of the sections has left me a bit uncertain, as I'm not entirely sure of your intended direction. I will further examine your feedback to better understand your point and make appropriate revisions. If you could provide additional guidance or specific suggestions, I would greatly appreciate it.

Comments 3:

please add some images to different classes of the dataset in the dataset description section

Response 3:

Thank you very much for your suggestion. I've followed up by incorporating additional supplementary materials into the manuscript, as you suggested. Specifically, I've included images that illustrate different classes of the dataset within the dataset description section. I firmly believe that these additions will greatly contribute to improving the clarity and comprehensibility of the manuscript. Your guidance is genuinely appreciated as we strive to enhance the quality of the work.

In addition, a description has been added to the manuscript as follows: " Different image categories are described in the supplementary material. " This addition appears on page 4, lines 20 to 21, which introduces the supplementary material.

Reviewer 3 Report

Comments and Suggestions for Authors

Thanks to the authors. All my comments were adequately addressed.

Author Response

We greatly appreciate your positive feedback and recommendation. We are thankful for your detailed review and are delighted that our work is deemed suitable for publication in the journal.